# Laser-Deposited Multilayer Coatings for Brake Discs: Corrosion Performance of 316L/430L Systems Reinforced with WC and TiC Particles

**DOI:** 10.3390/ma19010024

**Published:** 2025-12-20

**Authors:** Mohammad Masafi, Mo Li, Heinz Palkowski, Hadi Mozaffari-Jovein

**Affiliations:** 1Institute of Metallurgy, Clausthal University of Technology, Robert-Koch-Str. 42, D-38678 Clausthal-Zellerfeld, Germany; heinz.palkowski@tu-clausthal.de; 2Institute of Materials Science and Engineering Tuttlingen, Furtwangen University, Kronen-Str. 16, D-78532 Tuttlingen, Germany; mo.li@hs-furtwangen.de (M.L.); hadi.mozaffari@hs-furtwangen.de (H.M.-J.)

**Keywords:** laser metal deposition (LMD), corrosion, Microstructure, GJL brake discs, stainless steel 316L, stainless steel 430L, coating, reinforcement WC, reinforcement TiC

## Abstract

Grey cast iron brake discs are widely used in automotive applications due to their excellent thermal and mechanical properties. However, stricter environmental regulations such as Euro 7 demand improved surface durability to reduce particulate emissions and corrosion-related failures. This study evaluates multilayer coatings fabricated by Laser Metal Deposition (LMD) as a potential solution. Two multi-layer systems were investigated: 316L + (316L + WC) and 316L + (430L + TiC), which were primarily reinforced with ceramic additives to increase wear resistance, with their influence on corrosion being critically evaluated. Electrochemical tests in 5 wt.% NaCl solution (DIN 17475) revealed that the 316L + (316L + WC) coating exhibited the lowest corrosion current density and most stable passive behavior, consistent with the inherent passivation of the austenitic 316L matrix. In contrast, the 316L + (430L + TiC) system showed localized corrosion associated with micro-galvanic interactions, despite the chemical stability of TiC particles. Post-corrosion SEM and EDS confirmed chromium depletion and chloride accumulation at corroded sites, while WC particles exhibited partial dissolution. These findings highlight that ceramic reinforcements do not inherently improve corrosion resistance and may introduce localized degradation mechanisms. Nevertheless, LMD-fabricated multilayer coatings demonstrate potential for extending brake disc service life, provided that matrix–reinforcement interactions are carefully optimized.

## 1. Introduction

Grey cast iron with lamellar graphite is widely used in the automotive industry, particularly for brake disc applications, due to its excellent thermal conductivity, high damping capacity, good castability, and cost-effectiveness [1]. These properties make it ideal for dissipating the intense heat generated during braking, while also minimizing noise and vibration under dynamic loading conditions [2]. Its machinability and mechanical robustness further support its long-standing use in mass production of brake systems [3].

For decades, grey cast iron has remained the standard material for brake discs in passenger and commercial vehicles. Its ability to withstand thermal shocks and mechanical stresses during repeated braking cycles has made it indispensable in automotive engineering [4]. However, despite its mechanical advantages, grey cast iron suffers from poor corrosion resistance and significant wear, especially under humid or saline conditions. These limitations result in the release of fine particulate matter during braking, which contributes to non-exhaust emissions [5,6].

The environmental impact of brake wear particles has become a growing concern. These particles, typically in the PM10 and PM2.5 range, are released into the atmosphere and can be inhaled by humans, leading to respiratory and cardiovascular diseases [7,8]. According to the European Environment Agency, exposure to fine particulate matter is responsible for hundreds of thousands of premature deaths annually in Europe [9]. In response, the upcoming Euro 7 regulations introduce strict limits on brake-related particulate emissions, setting a maximum of 7 mg/km, with further reductions expected in the future [10].

To meet these new environmental standards, the automotive industry is actively seeking innovative material solutions that reduce particulate emissions while maintaining or improving braking performance. One promising approach is the application of wear- and corrosion-resistant coatings on grey cast iron brake discs. Among the various surface engineering techniques, laser metal deposition has emerged as a forward-looking technology for the development of next-generation brake disc materials [11].

Laser metal deposition is a directed energy deposition process in which a laser beam creates a melt pool on the substrate surface, into which metallic or composite powders are injected. The molten material solidifies rapidly, forming a dense, metallurgically bonded coating with minimal dilution of the substrate [12]. This process allows for precise control over the composition, thickness, and microstructure of the deposited layers, making it ideal for functionalizing the surface of grey cast iron components [13].

In contrast to selective laser melting (SLM), which is a powder bed fusion technique, laser metal deposition (LMD) employs a blown powder feed and a focused laser beam to create a melt pool on the substrate surface. While SLM offers high resolution and is widely used for near-net-shape manufacturing of complex components, it is limited by build size and requires full part fabrication. LMD, on the other hand, enables localized deposition, repair, and functional coating of existing components, with higher deposition rates and lower residual stresses due to reduced thermal gradients. These differences make LMD particularly suitable for surface functionalization of large parts such as grey cast iron brake discs, where dimensional integrity and metallurgical bonding are critical [14].

Compared to conventional coating methods such as thermal spraying or electroplating, laser metal deposition offers several advantages: superior adhesion due to metallurgical bonding, reduced porosity, high deposition rates, and the ability to process complex geometries [15]. These features make it particularly suitable for coating grey cast iron brake discs, which are otherwise difficult to modify due to their brittle graphite-rich microstructure.

In this study, two stainless steel alloys—316L and 430L—were selected as matrix materials for the coating system (CS). Stainless steel 316L is an austenitic alloy known for its excellent corrosion resistance in chloride-rich environments, attributed to its high chromium, nickel, and molybdenum content [16]. In contrast, stainless steel 430L is a ferritic alloy with lower nickel content, offering better thermal conductivity and cost-efficiency, but reduced corrosion resistance compared to 316L [17].

Ceramic particles were incorporated into the metallic matrices. While WC is well known for its exceptional hardness and wear resistance. Although TiC is chemically stable, its specific influence on the corrosion behavior of stainless steel has received comparatively little attention in the literature and remains underexplored [18,19]. The interaction between these particles and the surrounding matrix plays a critical role in determining the overall corrosion behavior of the CS.

However, the corrosion performance of these multilayer coatings, particularly under standardized testing conditions, remains underexplored. Given the increasing importance of corrosion resistance in the context of Euro 7 regulations, it is essential to evaluate the electrochemical stability of these coatings using established protocols. “Therefore, this study focuses on the microstructure and corrosion behavior of two multilayer coating systems produced by laser metal deposition: CS1, consisting of a 316L stainless steel layer reinforced with WC as the second layer, and CS2, consisting of a 430L stainless steel layer reinforced with TiC as the second layer. In both systems, the first layer is 316L stainless steel deposited on a grey cast iron (GJL) substrate.

The corrosion behavior of the coated samples was evaluated using electrochemical methods in accordance with the DIN EN ISO 17475 standard, which provides guidelines for potentiostatic and potentiodynamic polarization testing. This standard enables the assessment of localized corrosion, passivation behavior, and general electrochemical stability of metallic coatings in aggressive environments [20].

The aim of this work is to provide a comprehensive understanding of the corrosion mechanisms in multilayer coatings fabricated by laser metal deposition and to assess their potential as a future-oriented material solution for environmentally compliant and long-lasting brake disc applications.

## 2. Materials and Methods

### 2.1. Materials and Sample Preparation

The objective of this study was to investigate the feasibility of laser cladding grey cast iron (EN-GJL-150) brake disc (SHW Brake Systems GmbH, Tuttlingen, Germany) substrates with multilayer coatings based on stainless steel alloys 316L and 430L, reinforced with hard ceramic particles. Two distinct multilayer systems were designed and fabricated using laser metal deposition. Each system consisted of two layers: a corrosion-resistant base layer and a wear-resistant top layer.

Figure 1 depicts a three-dimensional representation of the resulting CS of the samples from top view. The first layer in both systems was composed of 316L stainless steel powder (20–53 µm, max. 5% oversize/undersize), selected for its excellent corrosion resistance and mechanical properties. The second layer of the first sample was reinforced with 30 to 40 wt.% spherical WC particles (5–40 µm) embedded in a 316L matrix. The second sample used a 430L stainless steel matrix (20–53 µm, max. 5% oversize/undersize) reinforced with 30 to 40 wt.% angular TiC particles (5–50 µm). All powders were supplied in gas-atomized form and used as received.

The coatings were deposited on grey cast iron brake disc substrates using a laser metal deposition system equipped with a coaxial powder nozzle and a continuous-wave fiber laser. The process parameters (laser power, scanning speed, powder feed rate, and shielding gas flow) were optimized based on preliminary trials to ensure good metallurgical bonding and minimal dilution. The laser power was set to 20 kW.

The samples were cut from the same”loca’Ion and top view of the coated brake disc, then embedded in epoxy resin, ground, and polished to a final surface finish of 1 µm using diamond suspension. After polishing and prior to corrosion testing, the surface roughness (Ra values) of all samples was measured using a Zeiss profilometer (Carl Zeiss AG, Oberkochen, Germany). Surface roughness is a critical parameter influencing corrosion behavior, and measurements were taken to ensure consistency across samples.

As illustrated in Table 1, the chemical compositions of the used materials are presented. The chemical composition of the gas atomized powders utilized for coating the samples was provided by the supplier, Höganäs (Bruksgatan 34, 263 83 Höganäs, Skåne County, Sweden).

The samples investigated are illustrated in Table 2. The samples were separated, embedded, and polished to evaluate their microstructure. The samples were mechanically polished to 1 µm to achieve a high-quality surface finish. For each CS (316L + (316L + WC) and 316L + (430L + TiC)), three samples were prepared. Surface roughness was measured three times on each sample using a Zeiss profilometer, resulting in a total of 18 measurements.

### 2.2. Microstructural Characterization

The microstructure of the coatings was analyzed using optical microscopy and scanning electron microscopy (SEM ZEISS EVO MA 15) (Carl Zeiss Industrielle Messtechnik GmbH (Oberkochen, Germany) before and after corrosion test. Energy-dispersive X-ray spectroscopy (EDS) (Bruker Nano GmbH, Berlin, Deutschland) was employed to evaluate the distribution of alloying elements and hard particles within the matrix. The interface between the coating and the substrate was also examined to assess bonding quality and dilution effects.

### 2.3. Corrosion Testing

Electrochemical corrosion tests were conducted in accordance with DIN EN ISO 17475 [20], a standardized method that provides detailed guidelines for performing potentiostatic and potentiodynamic polarization measurements on metallic materials. This standard is particularly suitable for evaluating the electrochemical kinetics of anodic and cathodic reactions, the onset of localized corrosion, and the repassivation behavior of metallic surfaces.

In summary, DIN EN ISO 17475 was selected because it offers a comprehensive, standardized, and scientifically validated approach to assess the corrosion behavior of complex, multilayered metallic coatings in saline environments.

Potentiodynamic polarization measurements were carried out in a 5 wt.% NaCl (Sodium chloride (Merck KgaA, Darmstadt, Germany) solution at room temperature using a three-electrode configuration: the coated sample served as the working electrode, a saturated calomel electrode (SCE) as the reference electrode, and a graphite rod as the counter electrode. The exposed area of the working electrode was approximately 1.5 cm^2^. Prior to testing, all samples were stabilized at open circuit potential (OCP) for about 120 min. The potential was scanned from −0.8 V to + 0.8 V versus SCE at a scan rate of 1 mV·s^−1^. The polarization resistance ($R_{P}$) was calculated using the Stern–Geary equation.R_p_ = β_a_ × β_c_/(2.3 × j_corr_ × (β_a_ + β_c_)(1)
where β_a_ and β_c_ are the anodic and cathodic Tafel slopes, and $j_{\mathrm{corr}}$ is the corrosion current density.

### 2.4. Data Availability and Ethics

All data generated or analyzed during this study are available from the corresponding author upon reasonable request. No human or animal subjects were involved in this research, and therefore no ethical approval was required.

### 2.5. Use of Generative AI

Generative artificial intelligence was used solely for language editing and formatting support. No AI tools were used for data generation, analysis, or interpretation.

## 3. Results

This section presents a concise and structured overview of the experimental findings, their interpretation, and the conclusions derived from the corrosion behavior of multilayer coatings. The samples were prepared as described in Section 2, using LMD to apply multilayer coatings on grey cast iron substrates. Prior to corrosion testing, surface morphology was examined using optical and SEM, and elemental composition was analyzed via EDS. Electrochemical corrosion tests were then conducted in a 5 wt.% NaCl solution. Post-corrosion analyses included repeated SEM and EDS investigations at selected corrosion sites to assess degradation mechanisms. Each subsection of the results is supported by comparative analysis with relevant literature to contextualize the findings.

### 3.1. Surface Morphology

The surface morphology of the multilayer coatings was examined prior to corrosion testing using optical microscopy and scanning electron microscopy (SEM). The top surfaces of both 316L + (316L + WC) and 316L + (430L + TiC) coatings exhibited dense and continuous structures with no visible cracks or delamination. SEM images revealed a relatively uniform distribution of hard particles within the stainless steel matrices. In particular, WC particles appeared well-embedded in the 316L matrix, while TiC particles in the 430L matrix showed slightly more pronounced surface protrusions.

The surface of the 316L + (316L + WC) coating appeared smoother and more homogeneous compared to the 316L + (430L + TiC) layer, which exhibited a slightly rougher texture due to the larger size and angular shape of the TiC particles. These morphological differences are consistent with previous studies on particle-reinforced stainless steel coatings fabricated by LMD [21].

No significant porosity was observed on the surface of either coating, indicating a high-quality deposition process. The good metallurgical bonding between the layers and the substrate suggests that the LMD parameters were well-optimized.

### 3.2. Microstructure

The microstructure of the multilayer coatings was analyzed using light microscopy and SEM imaging at the top of the deposited clad. As shown in Figure 2, the 316L + (316L + WC) layer exhibits a dense microstructure with uniformly distributed WC particles embedded in the austenitic matrix.

In contrast, Figure 3 illustrates the 316L + (430L + TiC) layer, where TiC particles are more angular and tend to align closely, sometimes forming chain-like arrangements. No significant porosity or delamination was observed in either coating, indicating a stable deposition process via LMD. These observations are consistent with previous studies on particle-reinforced stainless steel coatings fabricated by LMD [4]. Prior to corrosion testing, EDS analysis confirmed the presence and distribution of WC and TiC particles within their respective matrices. EDS analysis will be performed before and after corrosion testing to evaluate changes in the elemental composition of selected corrosion sites in Section 3.4.

Some WC and TiC particles appeared scratched or fractured due to mechanical stress during metallographic preparation, as similarly reported in previous studies on hard-particle-reinforced coatings [22,23]. Notably, the more frequent fragmentation of TiC particles may not only result from mechanical stress during metallographic preparation, but may also reflect their intrinsically lower wear resistance compared to WC. This observation is consistent with comparative studies on ceramic reinforcements, which report that WC exhibits superior hardness, toughness, and wear behavior under abrasive and impact conditions [24]. In particular, WC particles tend to maintain their integrity during mechanical processing, whereas TiC particles are more prone to microcracking and edge chipping, especially under localized stress conditions [25].

### 3.3. Surface Roughness

Surface roughness measurements were performed on each of the six samples (three per coating type), with three measurements taken per sample using a Zeiss profilometer.

Figure 4 illustrates the average surface roughness of both coatings, including error bars representing the standard deviation of the nine measurements per coating. The 316L + (316L + WC) coating exhibited an average surface roughness (Ra) of 0.09 ± 0.03 µm, while the 316L + (430L + TiC) coating showed a slightly higher roughness of 0.49 ± 0.03 µm.

These values suggest a more textured surface for the 316L + (430L + TiC) layer, potentially due to the angular morphology and distribution of TiC particles as described above.

The low standard deviations indicate a consistent surface finish after polishing, which reflects the uniformity of the post-processing rather than the intrinsic stability of the LMD process. Therefore, these values should not be interpreted as a direct measure of process stability.

Surface roughness plays a critical role in corrosion behavior, as it influences the effective surface area exposed to the corrosive medium and can promote localized corrosion initiation. Studies have shown that increased roughness may lead to higher corrosion rates due to the formation of micro-crevices and enhanced electrolyte retention [26].

### 3.4. Corrosion Behavior

The results revealed distinct differences in electrochemical performance between the two systems.

The electrochemical parameters discussed in this section are derived from the potentiodynamic polarization curves shown in Figure 5, which illustrate the corrosion behavior of both coating systems under identical test conditions.

The 316L + (316L + WC) coating demonstrated a more noble corrosion potential ($E_{\mathrm{corr}}\approx -611.0\;\mathrm{mV}$) and a significantly lower corrosion current density ($j_{\mathrm{corr}}\approx 7.39\times 10^{-7}\;{A/\mathrm{cm}}^{2}$), indicating superior corrosion resistance. This behavior is attributed to the austenitic nature of the 316L matrix, which is known for its excellent passivation capability in chloride-containing environments [16]. Two distinct passive regions were identified in the polarization curve. In the potentiodynamic polarization curves, the green shaded areas represent the passive regions where the coating systems exhibit stable passivation behavior, characterized by low current density and the formation of a protective oxide layer. In contrast, the pink shaded areas indicate the onset of localized corrosion, corresponding to the pitting potential (E_pit_), where the passive film breaks down and pitting corrosion initiates. The first region, between approximately −0.2 V and 0 V vs. Ag/AgCl/sat. KCl, is likely associated with WC oxidation, as tungsten-based oxides (WO_3_, W_2_O_5_) tend to form early but are relatively unstable in chloride-containing media. These oxides are porous, soluble, and non-self-healing, resulting in a higher passive current density compared to stainless steel. This behavior can be explained by the electrochemical reaction of WC in aqueous environments, where WC undergoes oxidative dissolution according to the reaction in Equation (2) [27]:WC + 5H_2_O → WO_3_ + CO_2_ + 10H^+^ + 10e^−^(2)

The formation of WO_3_ and release of CO_2_ indicate that WC is thermodynamically unstable under acidic or chloride-containing conditions, leading to degradation of the hard particles and weakening of the composite coating [28]. The second region, extending from +0.2 V to +0.6 V, corresponds to the passivation of the 316L matrix through the formation of a dense Cr_2_O_3_-based film, supported by Fe_2_O_3_ and MoO_3_, which provides more stable protection. The overall passive current density ranged from 10^−6^ to 10^−3^ A/cm^2^, with higher values observed in the second region due to partial WC dissolution near 0 V, which reduces film compactness and promotes localized attack [29].

In contrast, the 316L + (430L + TiC) coating exhibited a more negative corrosion potential ($E_{\mathrm{corr}}\approx -665.3\;\mathrm{mV}$) and a significantly higher corrosion current density ($j_{\mathrm{corr}}\approx 2.83\times 10^{-6}\;{A/\mathrm{cm}}^{2}$), indicating a higher corrosion rate and reduced passivation compared to the 316L + WC system.

The polarization curve revealed a narrow passive region between approximately –0.5 V and 0 V vs. Ag/AgCl/sat. KCl, with a passive current density ($j_{\mathrm{passive}}$) around $10^{-4}\;{A/\mathrm{cm}}^{2}$, indicating less stable passivation compared to CS1. The pitting potential ($E_{\mathrm{pit}}\approx 0\;V$) marks the onset of localized corrosion, likely influenced by the ferritic nature of the 430L matrix and microgalvanic interactions with TiC particles. Although TiC is chemically stable and does not undergo corrosion, its influence on the overall electrochemical behavior of the multilayer coating cannot be conclusively determined in this study, as no reference data for 430L stainless steel alone is available. Therefore, while TiC may contribute to passive film formation, the observed behavior under chloride exposure reflects the combined effect of the matrix and reinforcement rather than TiC alone.

The quantitative results of the electrochemical tests are summarized in Table 3, which includes corrosion potential ($E_{\mathrm{corr}}$), corrosion current density ($j_{\mathrm{corr}}$), anodic and cathodic Tafel slopes (βa, βc), and corrosion rates. The anodic and cathodic Tafel slopes (βa and βc) were applied to calculate the polarization resistance ($R_{P}$) using Equation (1), as the reactions occurred in the activation-controlled region. These slopes do not directly indicate passive film formation; instead, passivation tendencies can be inferred from the corrosion current density ($j_{\mathrm{corr}}$), $R_{P}$ values, and corrosion rates. A lower $j_{\mathrm{corr}}$ combined with a higher $R_{P}$ generally reflects easier passivation. Based on the averaged data, the 316L + WC coating exhibited a $j_{\mathrm{corr}}$ of ≈7.39 × 10^−7^ A/cm^2^ and an $R_{P}$ of approximately 37 kΩ·cm^2^**,** whereas the 316L + (430L + TiC) coating showed a higher $j_{\mathrm{corr}}$ of ≈2.83 × 10^−6^ A/cm^2^ and a lower $R_{P}$ of about 12 kΩ·cm^2^. This indicates that the WC-reinforced system provides superior corrosion resistance under uniform conditions. The calculated corrosion rates further support this trend, with 316L + WC at ≈8.58 µm/year (0.00858 mmpy) compared to ≈32.89 µm/year (0.03289 mmpy) for 316L + (430L + TiC). The poorer performance of the TiC-reinforced system can be attributed to localized degradation mechanisms such as pitting and microgalvanic coupling within the ferritic 430L phase, which accelerate material loss despite the presence of hard TiC particles. Additionally, the more negative $E_{\mathrm{corr}}$ (−665 mV vs. −611 mV) for the TiC system suggests a higher thermodynamic tendency toward corrosion, reinforcing the role of microstructural heterogeneity in its electrochemical behavior [30]. Therefore, the corrosion rate alone does not contradict the overall electrochemical indicators of passivation quality, which favor the 316L + (316L + WC) system.

Chi-squared values indicate acceptable to good fit quality for both datasets. The inclusion of Tafel slopes provides insight into electrochemical kinetics, while corrosion rates offer practical relevance for material degradation. Adding the standard deviation of $j_{\mathrm{corr}}$ enhances the statistical robustness of the comparison. These results confirm the superior passivation tendency of the austenitic 316L matrix and highlight the influence of ceramic reinforcement type on the electrochemical stability of multilayer coatings.

To provide a clearer quantitative context for the role of reinforcement particles, we have included reference data for bare grey cast iron (GJL) and monolithic stainless steels (316L and 430L) from the literature. These materials represent the substrate and single-layer coatings without ceramic reinforcement. Table 3 summarizes typical corrosion parameters reported for these references in chloride-containing environments, compared to the multilayer coatings investigated in this study.

The comparison confirms that bare GJL exhibits the highest corrosion rate (≈0.25 mm/year) and most negative corrosion potential (≈−650 mV), reflecting its poor resistance in NaCl solutions [31,32]. Monolithic 316L shows excellent passivation with $j_{\mathrm{corr}}$ values around 2 µA/cm^2^ and $E_{\mathrm{corr}}$ near −250 mV [33,34], while 430L stainless steel demonstrates intermediate performance but remains susceptible to localized attack [35].

Figure 5 presents the potentiodynamic polarization curves of the two coating systems (CS1: 316L + (316L + WC) and CS2: 316L + (430L + TiC)) measured in 5 wt.% NaCl solution. The curves reveal distinct electrochemical behaviors influenced by both matrix composition and the type of reinforcement particles.

The corrosion potential ($E_{\mathrm{corr}}$) of CS1 is more noble compared to CS2, indicating a higher resistance to initial corrosion. The corrosion current density ($j_{\mathrm{corr}}$) is significantly lower for CS1 (316L + WC), indicating a slower corrosion rate and enhanced passivation compared to CS2 (316L + (430L + TiC)).

This improvement can be attributed to the austenitic 316L matrix, which generally exhibits superior corrosion resistance in chloride-containing environments. In contrast, the ferritic 430L matrix in CS2 is more prone to localized attack and galvanic interactions, which explains its higher $j_{\mathrm{corr}}$ and corrosion rate. Literature reports that $E_{\mathrm{corr}}$ for 316L typically ranges between −100 mV and + 100 mV [33], while for 430L it lies between −500 mV and −300 mV, confirming the inherent difference in corrosion behavior [34]. Although direct measurements of the individual base materials were not performed in this study, the observed shift of $E_{\mathrm{corr}}$ for CS1 to more negative values (≈−611 mV) suggests that the incorporation of WC particles may have influenced the electrochemical response, consistent with findings in previous studies. This negative shift does not negate the overall improvement in corrosion resistance, as indicated by the significantly lower $j_{\mathrm{corr}}$ and higher $R_{P}$ for CS1. The corrosion resistance of WC-based coatings varies significantly depending on the deposition technique. According to Ward et al. (2011) [35], HVOF-sprayed WC cermet coatings on ferritic stainless steel exhibit poor corrosion performance in salt spray environments, primarily due to high porosity, microcracks, galvanic interactions, and phase dissolution, which promote substrate degradation [36]. In contrast, Mertgenç et al. (2023) [36] demonstrated that WC and TiC coatings applied via Electro-Spark Deposition (ESD) on high-speed steels significantly enhance both hardness and corrosion resistance. Notably, TiC coatings showed up to a threefold improvement in corrosion resistance, making them particularly suitable for applications in aggressive environments [37].

CS2 exhibits a passivation region, but the passivation current density j_p_ is higher, indicating less stable passive film formation. The pitting potential E_pit_ of CS2 is lower, which correlates with the presence of microgalvanic cells between 430L and TiC, leading to localized corrosion sites Similar behavior was reported by Haoming and Dejun [37], who observed that TiC-reinforced Fe-based coatings produced by laser cladding showed enhanced electrochemical resistance with increasing TiC content, yet also revealed localized corrosion phenomena attributed to interfacial reactions and microstructural heterogeneity 

Figure 6 presents light microscopy and SEM images of the CS1 coating system (316L + (316L + WC)), highlighting the microstructural features before and after corrosion testing. The micrographs clearly illustrate the corrosion-induced degradation of the CS1 coating system (316L + (316L + WC)). Prior to corrosion, the spherical WC particles appear well-embedded within the austenitic 316L matrix, showing smooth interfaces and minimal surface irregularities. After exposure to the corrosive environment, the WC particles exhibit pronounced surface roughening and partial dissolution, indicating their susceptibility to chemical attack [38].

This observation supports the electrochemical findings, where CS1 showed a relatively stable passive region but also signs of localized corrosion. The matrix itself displays mild corrosion effects, particularly near the particle-matrix interfaces. These regions may correspond to areas of microgalvanic interaction, where the electrochemical potential difference between WC and the surrounding austenitic steel promotes localized degradation [28].

The corrosion behavior of the matrix may also be influenced by the austenitic grain boundary characteristics, which are known to affect passivation stability. The degradation pattern suggests that WC, despite its mechanical advantages, may compromise corrosion resistance when not uniformly protected by the passive film.

Post-corrosion SEM images reveal pronounced surface roughening and partial dissolution of WC, indicating their susceptibility to chemical attack. These findings are consistent with Huang et al. [39]. who reported partial melting and degradation of WC particles in laser-cladded coatings. Additionally, Revilla and De Graeve (2022) identified particle-matrix interfaces as critical sites for corrosion initiation, supporting the observed localized damage in CS1 [40].

In SEM Figure 7, a WC particle from the CS1 coating system (316L + (316L + WC)) after corrosion testing in 5 wt.% NaCl solution is shown. The image highlights white interfacial lines surrounding WC particles, marked by yellow arrows, which indicate elemental diffusion during the laser deposition process. This observation suggests that W and C may have partially diffused from the WC particle into the surrounding 316L matrix during the laser deposition process. The white arrows in Figure 7 indicate the outward diffusion of W and C from the WC particle, while the black curved arrows represent the interaction of Cr from the 316L matrix with the diffused C. Based on literature and thermodynamic considerations, these interactions can lead to the formation of secondary carbides such as CrC at the interface. Although these regions contribute to mechanical reinforcement, their heterogeneous composition and disrupted passive film may act as preferential sites for corrosion initiation [39]. Similarly, the formation of intermetallic phases within the Fe–W–C ternary system—most likely the τ_1_-phase (Fe_3_W_3_C or Fe_7_W_6_)—is hypothesized based on the phase diagram and previous findings by Masafi et al. (2023) [4]. EDS analysis near fractured WC particles indicates W diffusion into the surrounding matrix, which may promote the development of these intermetallic phases. These newly formed carbides are electrochemically more noble than the surrounding steel matrix, which promotes microgalvanic coupling and localized corrosion [28]. In addition, WC itself is thermodynamically unstable in chloride-containing environments and may undergo oxidative dissolution according to Equation (2), forming WO_3_ and releasing CO_2_ [41]. This mechanism, combined with possible galvanic interactions involving τ_1_-phase carbides, could explain localized attack near reinforcement particles. These hypotheses require further confirmation through high-resolution phase analysis (e.g., EBSD or XRD) in future work.

Energy-dispersive X-ray spectroscopy (EDS) was performed at selected points on the CS1 coating system before and after corrosion testing, as shown in Figure 8 and Table 4. Spectra points S1 and S2 correspond to measurements on the matrix prior to corrosion, while S3 and S4 were taken after corrosion. The results indicate a slight but consistent reduction in Chromium (Cr) content, from approximately 17.7% (S1–S2) to 16.5% (S3–S4), which represents a relative decrease of about 6–7%. Although the absolute difference appears small, such localized Cr depletion can be critical for passive film stability, particularly in stainless steels exposed to sensitization temperatures (450–850 °C), where Cr-rich carbides (Cr_23_C_6_) may form at grain boundaries. This phenomenon weakens the protective oxide layer and increases susceptibility to intergranular corrosion. While the measured Cr values remain above the critical threshold of 12%, the observed trend supports the hypothesis of elemental redistribution and passive film disruption during corrosion. This supports the hypothesis of Cr depletion due to elemental diffusion and passive film disruption, as discussed in Figure 7.

Spectra points S5 and S6 represent measurements on WC particles prior to corrosion, while S7 and S8 were taken post-corrosion. The presence of Chlorine (Cl) and Sodium (Na) on WC particles after corrosion (e.g., S7) provides direct evidence of chloride-induced corrosion. This observation aligns with findings by Huang et al. (2023), who reported similar Cl and Na accumulation on WC surfaces after exposure to NaCl environments [39].

These results confirm that WC particles, while mechanically beneficial, are vulnerable to localized corrosion in chloride-rich environments, especially when embedded in austenitic matrices.

Additionally, the diffusion process contributes to the partial decomposition of WC particles within the coating matrix. Under continuous-wave laser deposition conditions, secondary carbides are formed as a result of WC dissolution Ruiz-Luna et al., 2024 [42]. While this transformation may enhance local hardness and wear resistance, it can simultaneously reduce corrosion resistance due to Cr consumption and matrix destabilization. The released tungsten (W) and carbon (C) atoms interact with chromium from the 316L matrix, leading to the formation of chromium carbides such as CrC, which alter the microstructure and electrochemical behavior of the coating [42]. Secondary carbides formed in the coatings under continuous-wave mode, while partial WC dissolution improved wear resistance. These microstructural changes can influence corrosion by altering elemental distribution. EDS analysis (Table 4) indicates a slight Cr reduction after corrosion (about 6–7% relative), consistent with early sensitization stages. This trend is consistent with sensitization phenomena in stainless steels, where thermal exposure promotes chromium carbide precipitation at grain boundaries, reducing local Cr availability for passive film formation and increasing susceptibility to corrosion. Although EDS alone does not confirm carbide precipitation, it provides supportive evidence when combined with SEM observations and thermodynamic considerations [43].

Before and after-corrosion SEM imagings (Figure 9) revealed surface degradation in coating systems 316L + (430L + TiC). In this multilayer system, more pronounced damage was observed, including localized features resembling pitting corrosion. These features are likely caused by microgalvanic interactions between the ferritic 430L matrix and the TiC particles, which exhibit significant differences in electrochemical potential. The observed pitting corrosion in CS2 may be attributed to chromium depletion at grain boundaries, as ferritic stainless steels like 430L are known to form some carbides under thermal exposure, which locally reduce corrosion resistance [44]. The incorporation of TiC particles in the 430L matrix promotes acicular ferrite formation and grain refinement, which can improve mechanical stability. Nevertheless, the potential for microgalvanic interactions between TiC and the ferritic matrix may contribute to localized corrosion phenomena [45].

Figure 10 shows the microstructural characteristics of the 430L + TiC coating system after corrosion testing. The SEM images reveal localized pits predominantly in the ferritic matrix, but these pits frequently occur in close proximity to TiC particles, as highlighted in the low-magnification image. This spatial correlation suggests that microgalvanic effects cannot be fully excluded, even if TiC particles themselves remain intact due to their higher corrosion resistance. Literature reports indicate that during LMD processing, partial Ti diffusion from TiC can lead to the formation of fine eutectic phases near the matrix–particle interface, which may locally alter electrochemical behavior [19]. Such interfaces could act as preferential sites for corrosion initiation, consistent with the observed pit distribution.

The schematic diagram (right) complements the SEM findings by highlighting areas of material loss and corrosion product accumulation. The dashed orange line marks the clear surface of the schematic representation on the right and indicates damage caused by corrosion-related deterioration. Yellow-highlighted zones show regions affected by corrosion, while the blue arrows (both light and dark) pointto filled areas, suggesting partial re-deposition or accumulation of corrosion products, including localized pitting.

This combined visualization supports the interpretation of corrosion mechanisms and the microstructural evolution within the coating system, emphasizing the role of TiC particles in both electrochemical behavior and surface stability.

Figure 11 and Table 5 present the EDS analysis results of the 430L + TiC coating system before (left) and after (right) corrosion testing. The SEM micrographs in Figure 11 show the surface region at low magnification, with marked points (S1–S8) indicating the locations of EDS measurements.

Table 5 summarizes the atomic concentrations of key elements at selected points. Spectra S1 and S2 (before corrosion) compared to S6–S8 (after corrosion) reveal increased sodium (Na) and chlorine (Cl) levels, indicating accumulation of corrosion products from the NaCl environment [46]. A notable observation is the reduction in chromium (Cr) at corroded sites (S6–S8), supporting the hypothesis of localized chromium depletion and microgalvanic interactions, particularly near grain boundaries or TiC particles [47] At S2, the presence of Ti suggests diffusion from TiC into the 430L matrix during laser metal deposition, likely forming intermetallic phases within the iron-based matrix [4]. Comparison of TiC before and after corrosion (S3 vs. S4/S5) shows elevated carbon (~18%) and stable Ti content, confirming TiC chemical stability despite corrosion exposure. SEM images (Figure 11) reveal surface degradation in the 316L + (430L + TiC) system, including localized pitting-like features, possibly linked to electrochemical potential differences between the ferritic 430L matrix and TiC particles. Point EDS analysis (Table 5) corroborates reduced Cr and Na/Cl enrichment at corroded sites, consistent with localized attack. While these findings support chromium depletion and microgalvanic effects, definitive proof would require high-resolution line-scan EDS or EBSD to confirm elemental redistribution and carbide stability at particle–matrix interfaces. The absence of significant Ti changes after corrosion suggests TiC stability, whereas Ti diffusion during LMD (S2 vs. S4/S5) indicates interfacial phase formation during processing rather than corrosion-induced alteration.

Post-corrosion SEM imaging revealed surface degradation in both coating systems. In the 316L + (430L + TiC) system, more pronounced damage was observed, including localized features resembling pitting corrosion, likely caused by microgalvanic interactions between the ferritic 430L matrix and the TiC particles. EDS analysis performed before and after corrosion testing revealed localized elemental variations at selected corrosion sites, such as slight Cr depletion and chloride enrichment, which support the electrochemical findings. Notably, TiC particles remained chemically stable after corrosion testing, showing no visible degradation in SEM images. Their stability suggests that TiC does not act as an active corrosion site, which is consistent with its known high corrosion resistance. However, a direct comparison with the substrate would be required to quantify any contribution to passive film formation. In contrast, WC particles in the 316L + (316L + WC) system showed signs of degradation, particularly at the particle–matrix interface, indicating susceptibility to corrosion despite the overall stability of the austenitic matrix [48].

## 4. Discussion

The comparative corrosion performance of the multilayer coating systems—316L + (316L + WC) (CS1) and 316L + (430L + TiC) (CS2)—reveals distinct electrochemical and microstructural behaviors, primarily governed by matrix composition, ceramic reinforcement, and surface characteristics.

Influence of Matrix Composition

CS1 demonstrates superior corrosion resistance due to the austenitic nature of the 316L matrix, which forms a stable passive film in chloride environments. This aligns with literature reporting that 316L stainless steel maintains passivity under aggressive conditions owing to its chromium-rich oxide layer [49]. In contrast, CS2 incorporates ferritic 430L, which exhibits a more negative corrosion potential and higher current density, indicating reduced passivation. Ferritic stainless steels are reported to be more susceptible to chromium depletion and intergranular corrosion under thermal or chloride exposure [50]. In this study, point EDS suggests localized Cr reduction at corroded sites, consistent with these mechanisms, although further line-scan analysis would be required for confirmation.

Role of Ceramic Reinforcements

WC particles in CS1 enhance mechanical strength but exhibit partial dissolution and elemental redistribution after corrosion, which may influence passive film integrity through secondary carbide formation [42]. Although partial dissolution of WC and formation of tungsten oxides (WO_3_, W_2_O_5_) occur under chloride exposure, the overall corrosion resistance remains superior due to the stable passive film formed by the austenitic 316L matrix. Compared to monolithic 316L stainless steel reported in literature ($E_{\mathrm{corr}}$ ≈ −250 mV, $j_{\mathrm{corr}}$ ≈ 2.00 × 10^−6^ A/cm^2^, corrosion rate ≈ 2.00 × 10^−2^ mmpy), CS1 achieves a significantly lower $j_{\mathrm{corr}}$ (7.39 × 10^−7^ A/cm^2^) and corrosion rate (8.60 × 10^−3^ mmpy), confirming the beneficial effect of WC reinforcement combined with an austenitic matrix [33,34].

TiC particles in CS2, on the other hand, remain chemically stable and may assist in passive film formation, as TiO_2_ formation can enhance chromia adhesion [51]. However, TiC also introduces microgalvanic effects when embedded in ferritic matrices. Literature suggests that TiC particles can act as cathodic sites, potentially accelerating localized corrosion in the surrounding matrix [29]. This observation is consistent with the pitting and filler accumulation noted in CS2, although further evidence would be required to confirm this mechanism. TiC particles in CS2 remain chemically stable and do not exhibit visible degradation in SEM images; however, corrosion pits frequently occur near TiC inclusions, suggesting local electrochemical heterogeneity. Reported studies indicate that Ti diffusion from TiC during LMD can lead to eutectic phase formation at the matrix–particle interface, which may alter local corrosion behavior [19]. While TiC itself is highly corrosion-resistant, these interfaces are hypothesized to act as preferential sites for pitting initiation, consistent with mechanisms described for similar systems. Compared to monolithic 430L stainless steel reported in the literature [35] ($E_{\mathrm{corr}}$ ≈ −450 mV, $j_{\mathrm{corr}}$ ≈ 8.00 × 10^−6^ A/cm^2^, corrosion rate ≈ 6.00 × 10^−2^ mmpy), the multilayer coating CS2 (316L + (430L + TiC)) achieves a noticeable improvement, reducing the corrosion rate to 3.29 × 10^−2^ mmpy and $j_{\mathrm{corr}}$ to 2.83 × 10^−6^ A/cm^2^. This indicates that TiC reinforcement provides some benefit, likely through localized TiO_2_ formation, but does not fully overcome the inherent susceptibility of the ferritic 430L matrix to pitting and microgalvanic attack.

Benchmarking Electrochemical Performance

To contextualize these findings, Table 3 and the calculated $R_{P}$ values were benchmarked against representative literature data. The 316L + (316L + WC) coating (CS1) exhibited $E_{\mathrm{corr}}$ ≈ −0.61 V, $j_{\mathrm{corr}}$ ≈ 7.39 × 10^−7^ A/cm^2^, and $R_{P}$ ≈ 37.1 kΩ·cm^2^, outperforming monolithic 316L stainless steel ($R_{P}$ ≈ 13.0 kΩ·cm^2^, $j_{\mathrm{corr}}$ ≈ 2.00 × 10^−6^ A/cm^2^). The 316L + (430L + TiC) coating (CS2) achieved $R_{P}$ ≈ 12.4 kΩ·cm^2^ and $j_{\mathrm{corr}}$ ≈ 2.83 × 10^−6^ A/cm^2^, which is significantly better than ferritic 430L ($R_{P}$ ≈ 3.25 kΩ·cm^2^, $j_{\mathrm{corr}}$ ≈ 8.00 × 10^−6^ A/cm^2^) but still less stable than CS1. Both multilayer systems provide a substantial improvement over bare gray cast iron ($R_{P}$ ≈ 0.52 kΩ·cm^2^, $j_{\mathrm{corr}}$ ≈ 5.00 × 10^−5^ A/cm^2^), which typically exhibits corrosion rates above 0.25 mm/year. The calculated corrosion rates for CS1 (≈0.0086 mm/year) and CS2 (≈0.0329 mm/year) confirm that WC reinforcement combined with an austenitic matrix offers the most reliable corrosion protection, while TiC reinforcement in ferritic matrices provides moderate improvement but remains susceptible to localized attack.

Surface Roughness and Corrosion

The higher surface roughness of CS2 (Ra ≈ 0.49 µm) compared to CS1 (Ra ≈ 0.09 µm) increases the effective surface area and promotes electrolyte retention, which can enhance localized corrosion [52]. This trend is reflected in the electrochemical results: CS2, with the rougher surface, exhibited a higher $j_{\mathrm{corr}}$ (2.83 × 10^−6^ A/cm^2^) and lower $R_{P}$ (12.4 kΩ·cm^2^) than CS1 ($j_{\mathrm{corr}}$ ≈ 7.39 × 10^−7^ A/cm^2^, $R_{P}$ ≈ 37.1 kΩ·cm^2^). These observations suggest that increased roughness may reduce passivation stability and accelerate localized attack, consistent with mechanisms reported in the literature [52].

Localized Elemental Variations

EDS analysis revealed slight but consistent Cr depletion (≈6–7% relative) at corroded sites in CS2, particularly near grain boundaries and TiC particles, supporting the hypothesis of sensitization and passive film disruption [53]. Chloride enrichment detected in pits further confirms localized attack, while WC surfaces in CS1 also showed Na and Cl presence, indicating chloride-induced corrosion [54]. Although these changes do not alter the overall composition, they highlight local redistribution critical for corrosion initiation.

The proposed microstructural–corrosion relationships in this study are based on SEM/EDS observations and supported by literature rather than direct phase-resolved analysis. While the presence of chromium depletion near grain boundaries and localized attack adjacent to TiC inclusions suggests sensitization and microgalvanic effects, these interpretations should be considered hypotheses consistent with reported mechanisms for similar systems [29,55,56]. Future work incorporating EBSD, XRD, or high-resolution interfacial analysis would provide definitive evidence for phase transformations and carbide stability during corrosion exposure.

Overall Interpretation

When compared to bare grey cast iron (GJL), which typically exhibits corrosion rates around 2.50 × 10^−1^ mmpy and $j_{\mathrm{corr}}$ near 5.00 × 10^−5^ A/cm^2^, both multilayer systems offer a substantial improvement in corrosion resistance. CS1 (316L + (316L + WC)) delivers the most significant enhancement, reducing the corrosion rate by more than an order of magnitude relative to GJL and outperforming CS2 in all electrochemical indicators [31,32].

Although CS2 exhibited a lower calculated average corrosion rate, this parameter does not fully reflect localized degradation mechanisms such as pitting and interface-driven attack. Electrochemical indicators, including corrosion potential, current density, and passive film stability, consistently favor CS1. Therefore, coatings based on 316L with WC reinforcement provide more reliable corrosion protection under chloride exposure [57]. In contrast, 430L + TiC systems, despite the inherent chemical stability of TiC particles, are more susceptible to localized corrosion due to matrix sensitization and microstructural heterogeneity at particle–matrix interfaces [58,59].

Whether the potentiodynamic polarization curves and $R_{P}$ values provide useful insights into the corrosion behavior and passivation tendency of the multilayer coatings, these results should be interpreted as indicative rather than conclusive. Potentiodynamic tests primarily capture short-term electrochemical kinetics under dynamic conditions, which may not fully represent long-term passivation stability or localized corrosion phenomena. Complementary techniques such as electrochemical impedance spectroscopy (EIS) or open-circuit potential (OCP) monitoring over time are commonly employed to validate passive film integrity and detect early signs of pitting [60,61,62]. In the absence of such additional measurements, our statements regarding passive film stability have been revised to emphasize relative trends rather than absolute conclusions. Literature reports confirm that EIS can reveal differences in charge-transfer resistance and film capacitance between austenitic and ferritic stainless steels, while OCP monitoring provides valuable information on potential fluctuations associated with localized attack [28,55,63]. Future work will incorporate these methods to strengthen the understanding of passivation mechanisms in multilayer coatings.

## 5. Conclusions

This study examined the corrosion behavior of multilayer coatings composed of 316L and 430L stainless steel matrices reinforced with WC and TiC particles, fabricated via laser metal deposition on grey cast iron substrates. The results demonstrate that both matrix composition and ceramic reinforcement type significantly affect electrochemical stability and localized corrosion mechanisms.

The 316L + (316L + WC) system (CS1) exhibited superior corrosion resistance, characterized by a more noble corrosion potential and lower current density. This performance is primarily attributed to the austenitic nature of the 316L matrix, which promotes stable passivation in chloride environments. However, post-corrosion analysis revealed partial WC degradation and elemental redistribution, indicating that long-term stability may be compromised under aggressive conditions. Polarization analysis revealed two distinct passive regions: an initial region likely associated with WC oxidation and a second region corresponding to the formation of a Cr-rich passive layer on the 316L matrix. This highlights the combined influence of reinforcement and matrix composition on passivation behavior

In contrast, the 316L + (430L + TiC) system (CS2) showed higher susceptibility to localized corrosion, including pitting near TiC particles and grain boundaries. While TiC remained chemically stable and did not exhibit visible degradation, its presence may influence local electrochemical behavior through interface effects, as supported by literature on Ti diffusion and eutectic phase formation during LMD. The ferritic 430L matrix demonstrated reduced passivation efficiency and slight chromium depletion at corroded sites, confirming its lower corrosion resistance in chloride-rich environments.

Surface roughness also played a critical role: CS2 exhibited a significantly higher Ra value than CS1, promoting electrolyte retention and micro-crevice formation, which accelerate localized attack. EDS analysis further revealed localized elemental variations—such as minor Cr depletion and chloride enrichment—rather than bulk composition changes, supporting the proposed corrosion mechanisms.

Overall, 316L + WC coatings provide better global corrosion protection, whereas 430L + TiC systems, despite the inherent stability of TiC, are more vulnerable due to matrix sensitization and interface-driven effects. These findings highlight the importance of optimizing both matrix selection and reinforcement distribution to enhance the durability of LMD-fabricated coatings in chloride environments.

## Figures and Tables

**Figure 1 materials-19-00024-f001:**
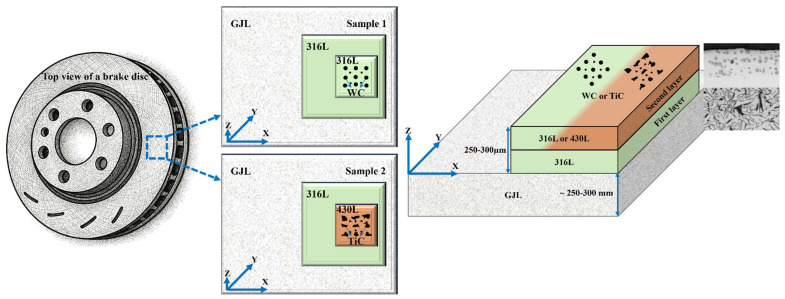
The production of the sample is shown schematically in three dimensions from a top view and Cross-section view. Two coatings are applied to the substrate: Sample 1: 316L, followed by 316L with spherical hard particles of WC, and Sample 2: 316L, followed by 430L with asymmetrical hard particles of TiC. The two coating layers are referred to as the first layer direct on substrate GJL and second layer direct on first Layer, respectively.

**Figure 2 materials-19-00024-f002:**
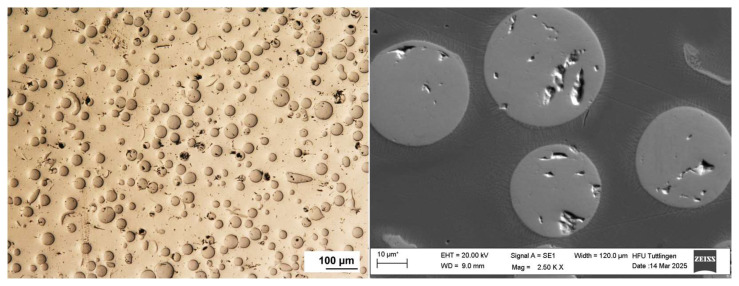
Top-view light (**left**) and SEM (**right**) image of the 316L + (316L + WC) coating layer of first sample showing a dense microstructure with uniformly distributed WC particles embedded in the austenitic 316L matrix. Some WC particles appear scratched due to mechanical stress during metallographic preparation. Note: The asterisk (*) next to the scale bar indicates that the scale was automatically generated by the ZEISS SEM system based on the magnification settings.

**Figure 3 materials-19-00024-f003:**
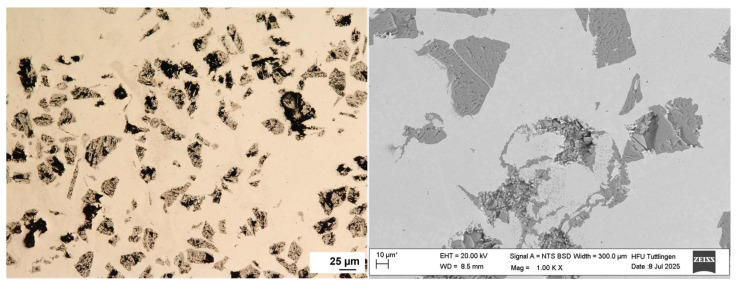
Top-view light (**left**) and SEM (**right**) image of the 316L + (430L + TiC) coating layer showing a dense microstructure with angular TiC particles aligned closely, occasionally forming chain-like structures within the ferritic 430L matrix. Some TiC particles broke due to mechanical stress during metallographic preparation. Note: The asterisk (*) next to the scale bar indicates that the scale was automatically generated by the ZEISS SEM system based on the magnification settings.

**Figure 4 materials-19-00024-f004:**
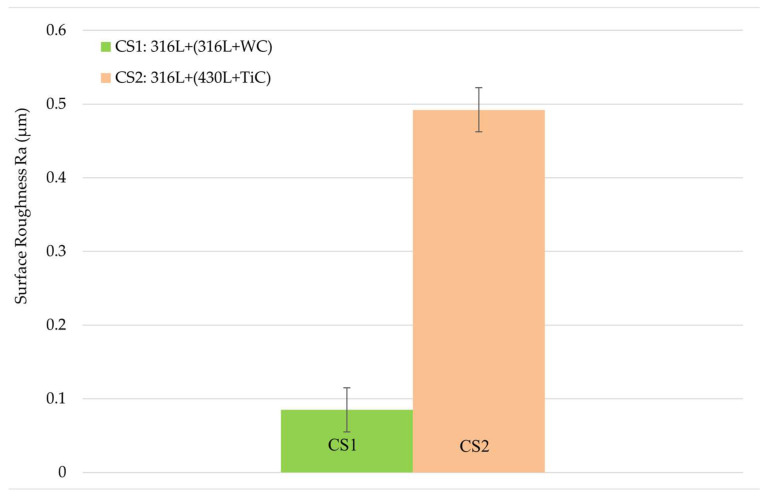
Average surface roughness (Ra) of the multilayer coatings, measured on three samples per coating type, with three measurements per sample. Error bars indicate the standard deviation of nine measurements per coating. The 316L + (430L + TiC) coating exhibits slightly higher roughness compared to the 316L + (316L + WC) coating, suggesting a more textured surface morphology due to the angular nature of TiC particles.

**Figure 5 materials-19-00024-f005:**
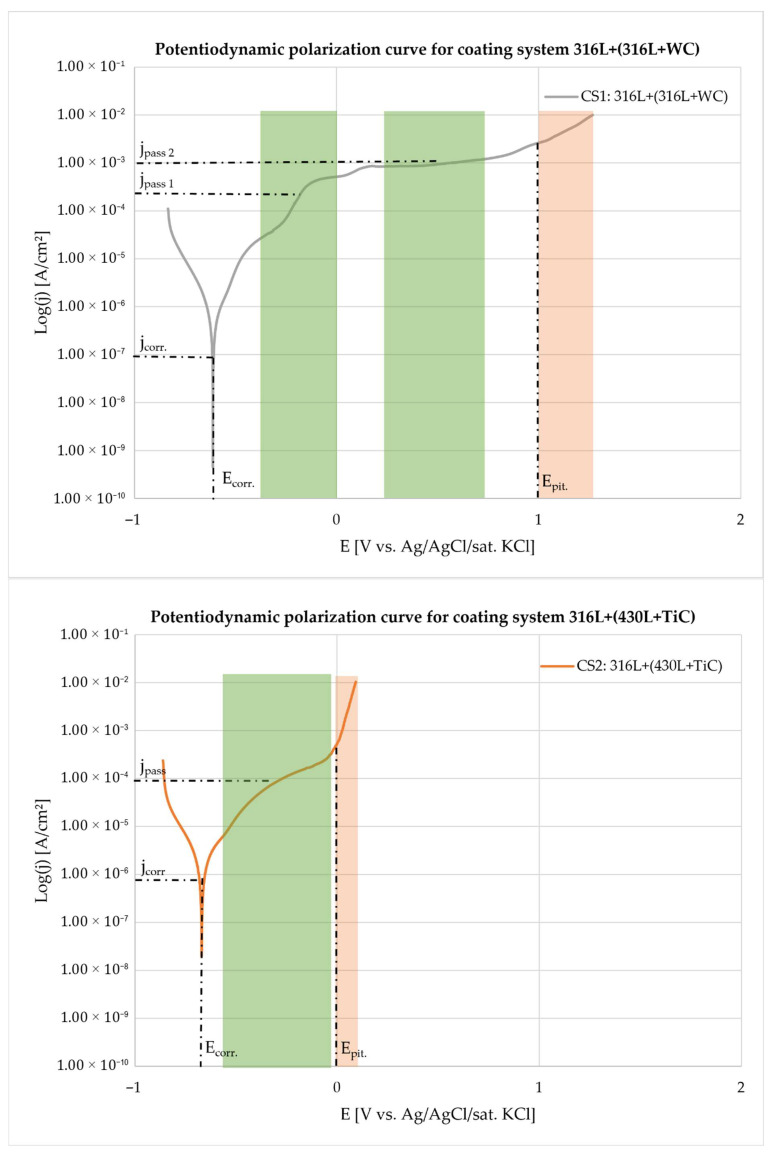
Potentiodynamic polarization curves of the multilayer coating systems 316L + (316L + WC) (CS1) and 316L + (430L + TiC) (CS2) measured in 5 wt.% NaCl solution. Each curve represents the average of three independent measurements per system. Key electrochemical parameters are highlighted, including corrosion potential ($E_{\mathrm{corr}}$), corrosion current density ($j_{\mathrm{corr}}$), passivation current density ($j_{\mathrm{passive}}$), and pitting potential ($E_{\mathrm{pit}}$). Green shaded regions indicate the passive domains where the coatings exhibit stable passivation behavior, while pink shaded regions mark the onset of pitting corrosion beyond the pitting potential. These areas enable direct comparison of passivation and localized corrosion resistance between the two coating systems.

**Figure 6 materials-19-00024-f006:**
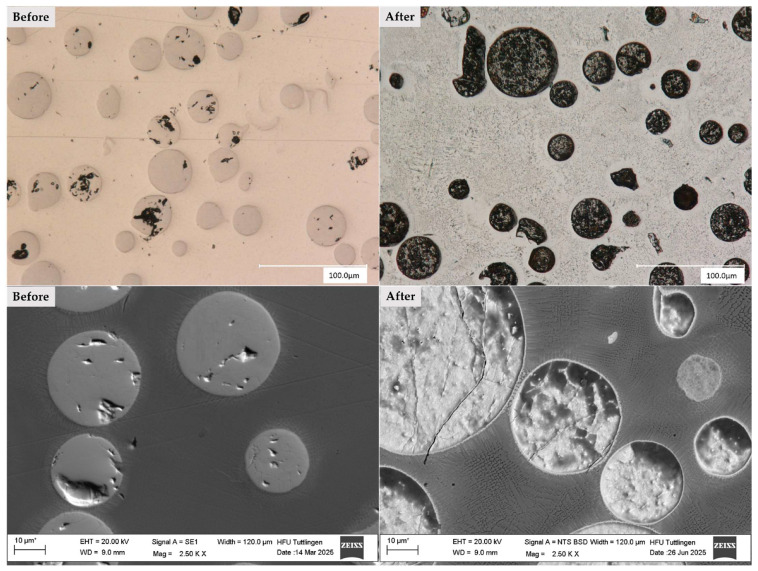
Light microscopy (**top row**) and scanning electron microscopy (**bottom row**) images of the CS1 coating system (316L + (316L + WC)) before (**left column**) and after (**right column**) corrosion testing in 5 wt.% NaCl solution. Prior to corrosion, the spherical WC particles appear well-embedded within the austenitic 316L matrix, showing smooth interfaces and minimal surface irregularities. After exposure to the corrosive environment, the WC particles exhibit pronounced surface roughening and partial dissolution, indicating their susceptibility to chemical attack. The surrounding matrix also shows mild corrosion effects, particularly near particle-matrix interfaces, suggesting localized degradation due to microgalvanic interactions. Note: The asterisk (*) next to the scale bar indicates that the scale was automatically generated by the ZEISS SEM system based on the magnification settings.

**Figure 7 materials-19-00024-f007:**
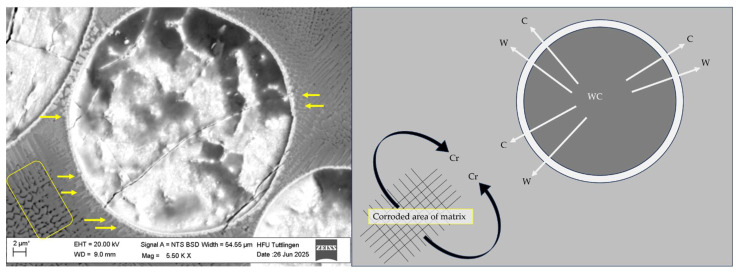
SEM image and schematic illustration of a dissolving WC particle in the CS1 coating system (316L + (316L + WC)) after corrosion testing in 5 wt.% NaCl solution. The SEM image (**left**) shows interfacial white lines around WC particles, marked by yellow line arrows, indicating elemental diffusion during the laser deposition process. The corroded area of the surrounding matrix is highlighted with a yellow rectangle. The schematic (**right**) visualizes the diffusion of W and C from the WC particle into the 316L matrix (indicated by white arrows) and the interaction of Cr from the 316L matrix with the diffused C from WC (indicated by black curved arrows), leading to the formation of secondary carbides such as CrC. These interfacial regions, while mechanically reinforcing, may act as weak points for corrosion initiation due to their heterogeneous composition and disrupted passive film formation. Note: The asterisk (*) next to the scale bar indicates that the scale was automatically generated by the ZEISS SEM system based on the magnification settings.

**Figure 8 materials-19-00024-f008:**
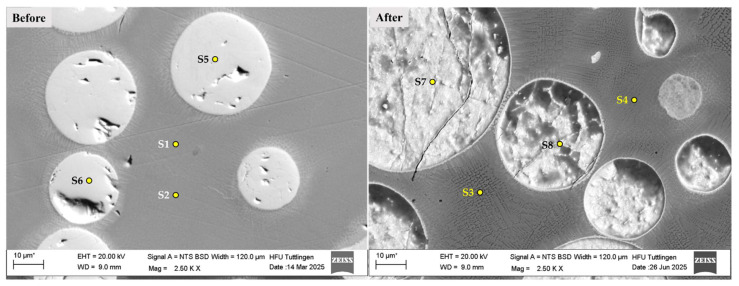
SEM images of the CS1 coating system before (**left**) and after (**right**) corrosion testing in 5 wt.% NaCl solution. Labeled points (S1–S8) indicate locations of EDS measurements. Post-corrosion images reveal surface degradation and elemental changes, particularly Cr reduction in the matrix and Cl/Na accumulation on WC particles. Note: The asterisk (*) next to the scale bar indicates that the scale was automatically generated by the ZEISS SEM system based on the magnification settings.

**Figure 9 materials-19-00024-f009:**
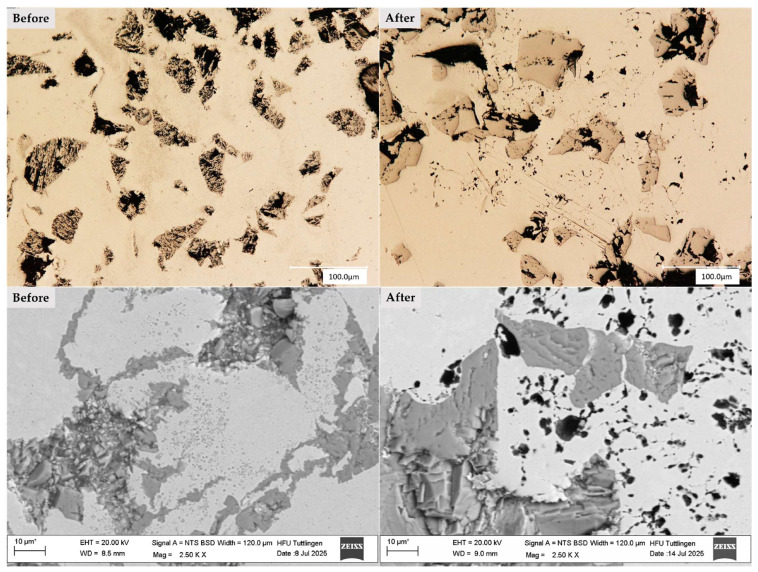
SEM images showing surface morphology before (**left**) and after (**right**) corrosion testing at two magnification levels. Top row: scale bar 100 µm; bottom row: scale bar 10 µm. Post-corrosion imaging reveals significant surface degradation, with localized pitting-like features attributed to microgalvanic interactions between the ferritic 430L matrix and TiC particles. TiC remained chemically stable and contributed to passive film formation. Note: The asterisk (*) next to the scale bar indicates that the scale was automatically generated by the ZEISS SEM system based on the magnification settings.

**Figure 10 materials-19-00024-f010:**
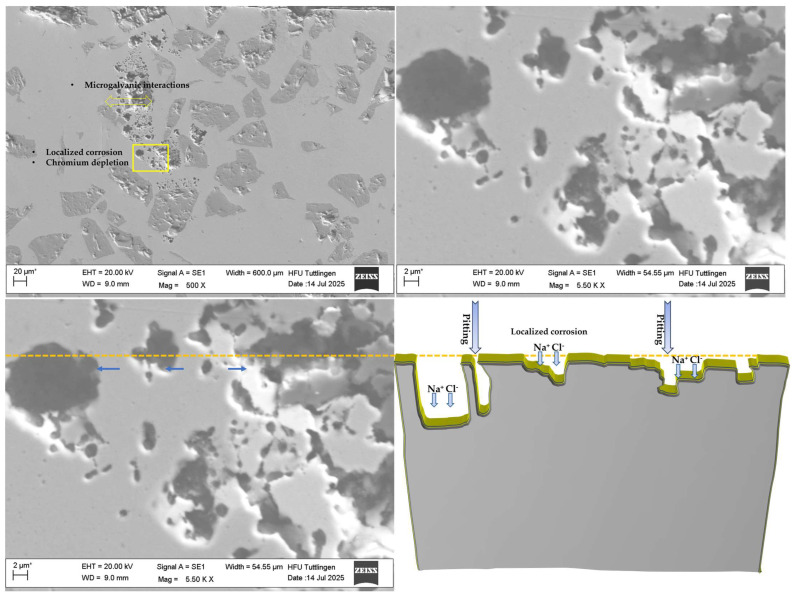
SEM image (**left**) and schematic diagram (**right**) of the 430L + TiC coating system after corrosion testing. SEM micrographs reveal cavities and localized pitting corrosion, likely associated with chromium depletion and microgalvanic interactions between the ferritic 430L matrix and TiC particles. The schematic highlights material loss and corrosion product accumulation (Na^+^/Cl^−^), with yellow contours indicating corrosion-affected regions and the dashed line marking the original surface level. Blue arrows (both light and dark) point to filled areas, suggesting partial re-deposition or accumulation of corrosion products, including localized pitting. These observations support localized attack mechanisms, while TiC particles remain chemically stable after corrosion exposure. Note: The asterisk (*) next to the scale bar indicates that the scale was automatically generated by the ZEISS SEM system based on the magnification settings.

**Figure 11 materials-19-00024-f011:**
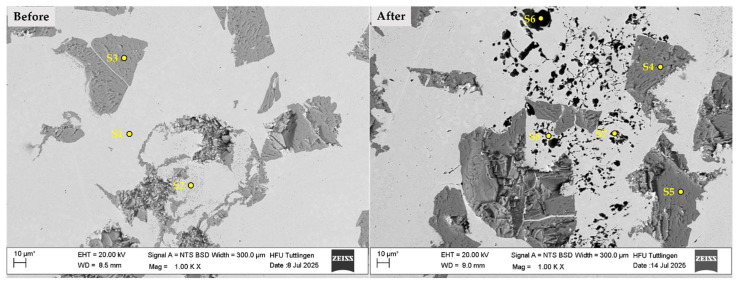
SEM micrographs of the 316L + (430L + TiC) coating system before (**left**) and after (**right**) corrosion testing. Marked points (S1–S8) indicate EDS measurement locations. Note: The asterisk (*) next to the scale bar indicates that the scale was automatically generated by the ZEISS SEM system based on the magnification settings.

**Table 1 materials-19-00024-t001:** Chemical composition of the GJL substrate, the 316L and 430L powder [3].

Element [wt.%]	GJL 150	316L	430L
C	3.50 ± 0.1	Max. 0.03	0.03
Si	2.00 ± 0.1	0.80	0.9
Mn	0.60 ± 0.05	1.0	0.1
P	<0.10 ± 0.02	-	0.01
S	<0.08± 0.02	<0.01	<0.01
Cu	0.20 ± 0.02	0.00	0.0
Cr	0.20 ± 0.02	17.00	17.00
Mo	0.35 ± 0.1	2.5	-
Ni	<0.20	12.00	<0.60
Sn	<0.10	-	-
N	-	-	-
Fe	Balance	Balance	Balance

**Table 2 materials-19-00024-t002:** List of coating systems.

Coating Systems (CS)	Substrate	First Layer	Second Layer	Hard Particles	Surface Condition
1	GJL	316L	316L	Spherical WC	1 µm polished
2	GJL	316L	430L	Angular TiC	1 µm polished

**Table 3 materials-19-00024-t003:** Electrochemical parameters derived from potentiodynamic polarization tests for the multilayer coating systems 316L + (316L + WC) (CS1) and 316L + (430L + TiC) (CS2) in 5 wt.% NaCl solution and bare GJL, monolithic 316L and 430 L. The table includes corrosion potential ($E_{\mathrm{corr}}$), corrosion current density ($j_{\mathrm{corr}}$) with standard deviation, anodic and cathodic Tafel slopes ($\beta_{a}$, $\beta_{c}$), calculated corrosion rates, and chi-squared values indicating the quality of the Tafel fit. Corrosion rates are expressed in micrometers per year (µm/year), equivalent to millimeters per year (mmpy), with 1 mmpy = 1000 µm/year. The inclusion of standard deviation for $j_{\mathrm{corr}}$ enhances the statistical robustness of the comparison between coating systems.

Coating System (CS) Material	E_corr_ [mV]	J_corr_ [A/cm^2^]	Standard Deviation	Beta A [V/dec]	Beta C [V/dec]	Corrosion Rate [mmpy]	Chi Squared	R_p_ [KΩ·cm^2^]
316L + (316L + WC) (1)	−611.0	7.39 × 10^−7^	5.07 × 10^−7^	1.30 × 10^−1^	1.23 × 10^−1^	0.86 × 10^−2^	10.91	37.060
316L + (430L + TiC) (2)	−665.3	2.83 × 10^−6^	1.19 × 10^−6^	1.92 × 10^−1^	1.39 × 10^−1^	3.29 × 10^−2^	9.46	12.357
Bare Gray Cast Iron [31,32]	−650.0	5.00 × 10^−5^	-	-	-	2.50 × 10^−1^	-	0.520
Monolithic 316L [33,34]	−250.0	2.00 × 10^−6^	-	-	-	2.00 × 10^−2^	-	13.000
Monolithic 430L [35]	−450.0	8.00 × 10^−6^	-	-	-	6.00 × 10^−2^	-	3.250

**Table 4 materials-19-00024-t004:** Atomic concentration percentages of selected elements (C, Na, S, Cl, Cr, Mn, Fe, Ni, W) at spectra points S1–S8 obtained by EDS analysis before and after corrosion testing. Notable changes include Cr depletion in the matrix and Cl/Na enrichment on WC particles.

Atomic Concentration [%]									
Spectrum	C	Na	Cl	Cr	Mn	Fe	Ni	Mo	W
S1	2.156	0.000	0.000	17.632	1.110	65.533	9.471	1.519	2.579
S2	1.781	0.000	0.000	17.694	1.178	65.852	9.447	1.512	2.536
S3	1.588	0.000	0.000	16.681	1.181	67.462	9.739	1.208	2.141
S4	1.552	0.000	0.000	16.473	1.216	67.382	9.915	1.249	2.214
S5	18.449	0.000	0.000	0.000	0.000	0.535	0.081	0.000	80.936
S6	15.694	0.000	0.000	0.051	0.087	0.803	0.620	0.000	82.745
S7	24.595	0.458	0.253	0.120	0.000	0.749	0.205	0.000	73.620
S8	27.738	0.646	0.690	0.441	0.000	0.720	0.261	0.000	69.504

**Table 5 materials-19-00024-t005:** Atomic concentrations [%] of selected elements at EDS measurement points before and after corrosion testing. Increased Na and Cl levels at S6, S7 and S8 indicate corrosion product accumulation. Reduced Cr content suggests localized chromium depletion. The absence of a significant change in TiC composition indicates chemical stability during corrosion.

Atomic Concentration [%]							
Spectrum	C	Na	Si	Cl	Ti	Cr	Fe
S1	1.315	0.000	0.677	0.000	1.878	16.398	79.732
S2	1.700	0.000	0.270	0.000	5.710	16.860	75.460
S3	18.213	0.000	0.122	0.000	80.829	0.094	0.742
S4	18.581	0.000	0.000	0.000	80.475	0.001	0.943
S5	18.169	0.000	0.000	0.000	81.215	0.000	0.616
S6	8.277	5.386	0.000	5.524	10.740	10.199	59.874
S7	7.347	3.807	0.000	1.819	15.350	15.332	56.345
S8	5.336	2.067	1.068	1.377	16.886	14.918	58.348

## Data Availability

The original contributions presented in this study are included in the article. Further inquiries can be directed to the corresponding author.

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
