# Peer review of "Laser-Deposited Multilayer Coatings for Brake Discs: Corrosion Performance of 316L/430L Systems Reinforced with WC and TiC Particles"

_materials, 2025, doi:10.3390/ma19010024_

Round 1
Reviewer 1 Report
Comments and Suggestions for Authors
The manuscript addresses a relevant topic and the corrosion study is generally solid, but several points could be clarified or expanded. The comments below are intended for improving clarity and impact.​
- The role of WC, TiC and the matrix alloys is discussed without direct reference samples (bare GJL, monolithic 316L/430L, or single‑layer coatings); including even one or two such references would make the impact of the particles much more quantitative.​
- Corrosion evaluation relies only on potentiodynamic polarization and Rp; if possible, adding one complementary method (e.g., EIS or simple immersion / OCP vs time) would strengthen statements about passivation stability and localized corrosion, or these claims could be phrased more cautiously.​
- The microstructural–corrosion link is mainly qualitative; phase‑resolved techniques (XRD, EBSD, higher‑resolution analysis near particles) would better support the proposed mechanisms, or the discussion could be explicitly framed as hypotheses consistent with literature.​
- Finally, the discussion could more clearly benchmark Ecorr, jcorr, Rp and corrosion rates against a few representative studies on 316L/430L and particle‑reinforced systems, to show where the present coatings stand relative to existing solutions.​
Author Response
In word attached

Reviewer 2 Report
Comments and Suggestions for Authors
This manuscript investigates the corrosion behavior of multilayer coatings consisting of 316L and 430L stainless steel matrices reinforced with WC and TiC particles, fabricated via laser metal deposition on grey cast iron substrates. However, some details in the methodology and analysis require further clarification. Below are specific comments for improvement.
- The authors should further enrich the ‘Introduction’ part to illustrate the differences among various of laser additive manufacturing techniques like selective laser melting (e.g., doi.org/10.1016/j.jma.2024.12.018) instead of simply describing the laser metal deposition technology.
- The discussion of dual passive regions in the 316L+(316L+WC) system is speculative. If two passive regions are claimed, the authors should provide direct evidence. Similarly, the conclusion that WC dissolution contributes to the second passive region needs stronger justification or should be toned down.
- The manuscript states that roughness influences corrosion behavior but does not quantitatively connect Ra values to electrochemical outcomes. The authors should either provide a clear correlation between roughness and corrosion parameters or separate this discussion from the electrochemical interpretation.
- Several statements attribute corrosion to ‘microgalvanic interactions’ or ‘chromium depletion’ without quantitative evidence. To support these claims, the authors should provide either line-scan EDS across.
- Scale bars should be clearly labeled (remove repeated notes about automatic scale generation). Schematic diagrams should be simplified to emphasize key mechanisms rather than repeating the microstructural view.
- The manuscript refers to the formation of CrC, τ1-phase carbides, and decomposition of WC, but no phase identification was performed. These statements must be clearly framed as hypotheses based on literature, not confirmed experimental observations in this study.
- The manuscript contains frequent grammatical errors, overly long sentences, repeated phrases, and ambiguous terminology. A thorough language revision is required to ensure clear scientific communication and improve readability.
Round 2
Reviewer 2 Report
Comments and Suggestions for Authors
The authors have addressed all questions I raised and I recommend it to be published in this state.